# A Method for Evaluating Hyperparameter Sensitivity in Reinforcement Learning

**Jacob  Adkins**
Department of Computing Science
University of Alberta; Amii
Edmonton, Canada
`jadkins@ualberta.ca`

**Michael  Bowling**
Department of Computing Science
University of Alberta; Amii
Edmonton, Canada
`mbowling@ualberta.ca`

**Adam  White**
Department of Computing Science
University of Alberta; Amii
Edmonton, Canada
`amw8@ualberta.ca`

## Abstract

The performance of modern reinforcement learning algorithms critically relies on tuning ever increasing numbers of hyperparameters. Often, small changes in a hyperparameter can lead to drastic changes in performance, and different environments require very different hyperparameter settings to achieve state-of-the-art performance reported in the literature. We currently lack a scalable and widely accepted approach to characterizing these complex interactions. This work proposes a new empirical methodology for studying, comparing, and quantifying the sensitivity of an algorithm's performance to hyperparameter tuning for a given set of environments. We then demonstrate the utility of this methodology by assessing the hyperparameter sensitivity of several commonly used normalization variants of PPO. The results suggest that several algorithmic performance improvements may, in fact, be a result of an increased reliance on hyperparameter tuning.

## 1   Introduction

The performance of reinforcement learning algorithms critically relies on the tuning of numerous hyperparameters. With the introduction of each new algorithm, the number of these critical hyperparameters continues to grow. Consider the progression of value-based reinforcement learning algorithms, starting from DQN (Mnih et al., 2015), which has 16 hyperparameters that the practitioner must choose, to Rainbow (Hessel et al., 2018) with 25 hyperparameters. This increase can be observed in Figure 1. This proliferation is problematic because performance can vary drastically with respect to hyperparameters across environments. Often, small changes in a hyperparameter can lead to drastic changes in performance, and different environments require very different hyperparameter settings to achieve the reported good performances (Franke et al., 2021; Eimer et al., 2022, 2023; Patterson et al., 2024). Generally speaking, hyperparameter tuning requires a combinatorial search and thus many published results are based on a mix of default hyperparameter settings and informal hand-tuning of key hyperparameters like the learning rate. Our standard evaluation methodologies do not reflect the sensitivity of performance to hyperparameter choices, and this is compounded by a lack of suitable metrics to characterize said sensitivities.

There are many different ways one could characterize performance with respect to hyperparameter choices in reinforcement learning, but the community lacks an agreed standard. Hyperparameter

38th Conference on Neural Information Processing Systems (NeurIPS 2024).

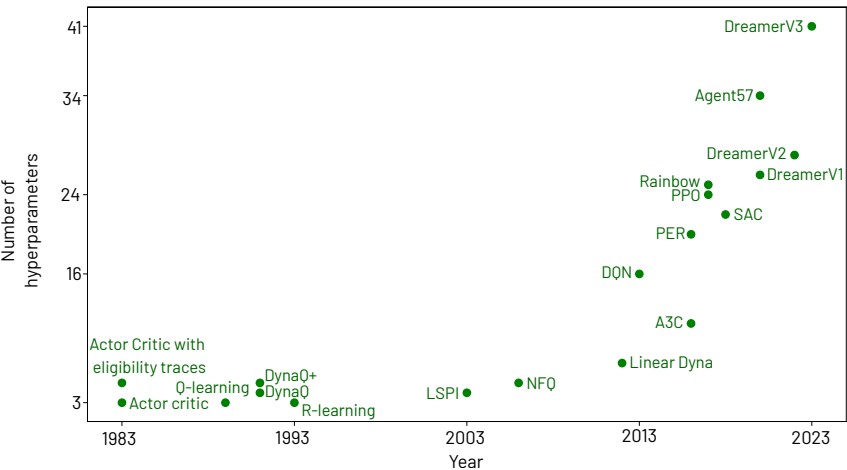

Figure 1: A count of hyperparameters for different reinforcement learning algorithms proposed over the last decade. We include value-based, policy-gradient, and model-based methods. The counts do not include hyperparameters controlling the network architectures, such as number of layers, activation functions, etc. See Appendix B for details on how hyperparameters were counted.

sensitivity curves, such as those found in the introductory textbook (Sutton & Barto, 2018), summarize performance with respect to several values of a key hyperparameter producing U-shaped curves. Sensitivity curves do not work well with many dimensions of hyperparameters, nor is there a well-established way to use them to summarize performance in multiple environments. If computation is of no object, then performance percentiles can be used to compute the likelihood that an algorithm will perform well if its hyperparameters are randomly sampled from some distribution (Jordan et al., 2020). Although very general, this approach does not reflect how practitioners tune their algorithms. The Cross-environment Hyperparameter Benchmark (Patterson et al., 2024) compares algorithms by a mean normalized performance score across environments but ultimately focuses on the possibility of finding a single good setting of an algorithm's hyperparameters that performs well rather than characterizing sensitivity. What performance-only metrics lack is a measurement of what proportion of realized performance is due to per-environment hyperparameter tuning.

We propose an empirical methodology to better understand the interplay between hyperparameter tuning and an reinforcement learning agent's performance. Our methodology consists of two metrics and graphical techniques for studying them. The first metric, called an algorithm's *hyperparameter sensitivity*, measures the degree to which an algorithm's peak reported performance relies upon per-environment hyperparameter tuning. This metric captures the degree to which per-environment tuning improves performance relative to the performance of the best-fixed hyperparameter setting across a distribution of environments. The second metric, named *effective hyperparameter dimensionality*, measures how many hyperparameters must be tuned to achieve near-peak performance. It is often unclear how important specific hyperparameters are and if they should be included in the tuning process. These two metrics can help us better understand existing algorithms and drive research toward algorithmic improvements that reduce hyperparameter sensitivity.

We validate the utility of our methodology by studying several variants of PPO (Schulman et al., 2017) that have been purported to reduce hyperparameter sensitivity and increase performance. We performed a large-scale hyperparameter study over variants of the PPO algorithm consisting of over 4.3 million runs (13 trillion environment steps) in the Brax MuJoCo domains (Freeman et al., 2021). We investigate the relationship between performance and hyperparameter sensitivity with several commonly used normalization variants paired with PPO. We found that normalization variants, which increased PPO's tuned performance, also increased sensitivity. Other normalization variants had negligible effects on performance and marginal effects on hyperparameter sensitivity. This result contrasts the view that normalization makes reinforcement learning algorithms easier to tune and, as a consequence, results in improved performance.

## 2 Problem Setting and Notation

We formalize the agent-environment interaction as a Markov Decision Process (MDP) with finite state space $\mathcal{S}$ and action space $A$, bounded reward function $\mathcal{R}: \mathcal{S} \times A \times \mathcal{S} \to \mathcal{R} \subset \mathbb{R}$, transition function $\mathcal{P}: \mathcal{S} \times A \times \mathcal{S} \to [0,1]$, and discount factor $\gamma \in [0,1]$. At each timestep $t$, the agent observes the state $S_t$, selects an action $A_t$, the environment outputs a scalar reward $R_{t+1}$ and transitions to a new state $S_{t+1}$. The agent's goal is to find a policy, $\pi: \mathcal{A} \times \mathcal{S} \to [0,1]$, that maximizes the expected return, $G_t \doteq R_{t+1} + \gamma R_{t+2} + ...$, in all states: $\mathbb{E}_\pi[G_t|S_t = s]$ for all $s \in \mathcal{S}$.

Most reinforcement learning agents learn and use approximate value functions in order to improve the policy through interaction with the world. The state-value function $v_\pi: \mathcal{S} \to \mathbb{R}$ is the state conditioned expected return following policy $\pi$ defined as $v_\pi(s) \doteq \mathbb{E}_\pi[G_t|S_t = s]$. Similarly, the action-value function $q_\pi: \mathcal{S} \times A \to \mathbb{R}$ provides the state and action conditioned expected return following policy $\pi$ defined by $q_\pi(s,a) \doteq \mathbb{E}_\pi[G_t|S_t = s, A_t = a]$. The advantage function $A_\pi(s,a) \doteq q_\pi(s,a) - v_\pi(s)$ describes how much better taking an action $a$ in state $s$ is rather than sampling an action according to $\pi(\cdot|s)$ and following policy $\pi$ afterward. Given an estimate of the value function $\hat{v}$, an agent can update estimates of the value of states based on estimates of the values of successor states. An $n$-step return is defined as $G_{t:t+n} \doteq R_{t+1} + \gamma R_{t+2} + \cdots + \gamma^{n-1} R_{t+n} + \gamma^n \hat{v}(S_{t+n})$. A mixture of $n$-step returns, called a truncated-$\lambda$ return can be created by weighting $n$-step returns by a factor $\lambda \in [0,1]$, $G_{t:t+n}^\lambda \doteq (1-\lambda) \sum_{j=1}^{n-1} \lambda^{j-1} G_{t:t+j} + \lambda^{n-1} G_{t:t+n}$. Truncated-$\lambda$ returns are often used in reinforcement learning algorithms such as PPO (Schulman et al., 2017) to estimate state values and state-action advantages, which are used to approximate the value function and improve the policy.

Of particular interest to this work is the setting where an empiricist is evaluating a set of algorithms $\Omega$ across a distribution of environments $\mathcal{E}$. Each algorithm $\omega \in \Omega$ is presumed to have some number of hyperparameters $n(\omega)$. Each hyperparameter $h_i$, $1 \le i \le n(\omega)$ is chosen from some set of choices $H_i^\omega$. The total hyperparameter space $H^\omega$ is defined as a Cartesian product of those choices $H^\omega \doteq H_1^\omega \times H_2^\omega \times ... H_{n(\omega)}^\omega$. Once an algorithm $\omega \in \Omega$ and a hyperparameter setting $h \in H^\omega$ are chosen, the tuple $(\omega, h)$ specifies an *agent*.

## 3 Hyperparameter Sensitivity

First, we must define what we mean by sensitivity. A *sensitive* algorithm is an algorithm that requires a great deal of per-environment hyperparameter tuning to obtain high performance. Conversely, an *insensitive* algorithm is one where there exist hyperparameter settings such that the algorithm can obtain high-performance across a distribution of environments with fixed hyperparameters.

This section presents two contributions: a metric for assessing an algorithm's hyperparameter sensitivity, and a method of graphically analyzing the relationship between hyperparameter sensitivity and performance along a 2-dimensional plane. These tools may be used to develop a deeper understanding of existing algorithms and we hope will aid researchers in evaluating algorithms more holistically along dimensions other than just benchmark performance.

### 3.1 Sensitivity Metric

We want a performance metric that summarizes the learning of an online reinforcement learning agent. The natural choice of performance metric is to report the average return obtained during learning, which we call the area under the (learning) curve (AUC). The AUC on a run is denoted by $p(\omega, e, h, \kappa)$ where $\omega \in \Omega$ is an algorithm, $e \in \mathcal{E}$ is an environment, $h \in H^\omega$ is a hyperparameter setting, and $\kappa \in \mathcal{K} \subset \mathbb{N}$ is the random number generator (RNG) seed. Performance observed during a run of a reinforcement learning agent depends on many factors: the reinforcement learning algorithm, the environment, the hyperparameter setting, and many forms of stochasticity. Even after fixing the algorithm, environment, and hyperparameter setting, performance distributions are often skewed and multi-modal (Patterson et al., 2023). Therefore, many runs are required to obtain accurate estimates of expected performance $\hat{p}(\omega, e, h) \doteq \frac{1}{|\mathcal{K}|} \sum_{\kappa \in \mathcal{K}} p(\omega, e, h, \kappa)$ where $\mathcal{K} \subset \mathbb{N}$ is the set of RNG seeds used during the experiment. In the experiments presented in this paper, we perform 200 runs averaging performance over a subset of runs, after filtering (described later), and report 95% bootstrap confidence intervals around computed statistics.

It is crucial to capture performance across sets of environments in order to compute sensitivity. Recall, that our notion of sensitivity captures the degree to which an algorithm relies on per-environment hyperparameter tuning for its reported performance gains. Our choice of AUC as a performance metric does not allow for cross-environment performance comparisons directly because the magnitudes of returns vary greatly between environments. Consider the distributions of performance presented in the left plot in Figure 2. The performance realized by good hyperparameter settings in Halfcheetah is orders of magnitude greater than the performance of good hyperparameter settings in Swimmer. Nevertheless, just because the absolute magnitude is lower or the range of observed performances is tighter, that does not mean the differences are any less significant. Thus, in order to consider how hyperparameters perform across sets of environments, we need to normalize performance to a standardized score.

In this work, we use $[5, 95]$ percentile normalization. We choose percentile normalization as it has a lower variance than alternatives like min-max normalization. Other normalization methods, such as min-max or CDF normalization (Jordan et al., 2020), could also be used with our hyperparameter sensitivity formulation. After conducting a large number of runs across different algorithms, environments, and hyperparameter settings, for each environment $e$, we find the 5th percentile $p_5(e)$ and 95th percentile $p_{95}(e)$ of the distribution of observed performance in $e$. Then, for each algorithm, environment, and hyperparameter setting, the *normalized environment score* is obtained by squashing performance:

$$\Gamma(\omega, e, h) \doteq \frac{\hat{p}(\omega, e, h) - p_5(e)}{p_{95}(e) - p_5(e)} \tag{1}$$

Note the right hand side of Figure 2, the distributions of normalized scores for hyperparameter settings in Swimmer and Halfcheetah now lie in a common range.

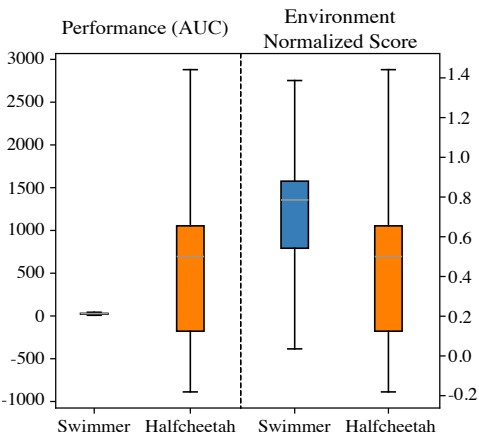

Figure 2: **Left**: The distributions of performance (AUC) over 625 hyperparameter settings for the PPO algorithm in Swimmer and Halfcheetah Brax environments. **Right**: The same distributions after applying score normalization. Each data point is the mean AUC across runs. Each run consisted of 3M steps of agent-environment interaction.

Normalized scores allow practitioners to determine which fixed hyperparameter settings do well across multiple environments. That is, a practitioner can find the hyperparameter setting that maximizes the mean normalized score across a distribution of environments. Consider the performance of the hyperparameter setting denoted by the blue stars in Figure 3. This setting performs in the top quartile of hyperparameter settings in both Swimmer and Halfcheetah. In contrast, consider the hyperparameter setting denoted by the red stars. While this hyperparameter setting sits near the top of the distribution for Halfcheetah, it performs poorly in Swimmer.

Given an algorithm $\omega \in \Omega$, we define its *hyperparameter sensitivity* $\Phi$ as follows:

$$\Phi(\omega) \doteq \frac{1}{|\mathcal{E}|} \sum_{e \in \mathcal{E}} \max_{h \in H^\omega} \Gamma(\omega, e, h) - \max_{h \in H^\omega} \frac{1}{|\mathcal{E}|} \sum_{e \in \mathcal{E}} \Gamma(\omega, e, h) \tag{2}$$

The hyperparameter sensitivity of an algorithm is the difference between its per-environment tuned score and its cross-environment tuned score. The *per-environment tuned score* is the average nor-

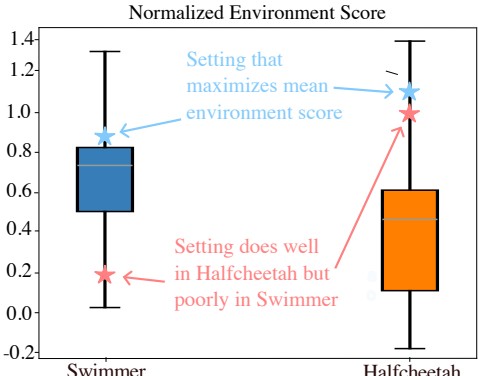

Figure 3: The distributions of environment normalized scores for 625 hyperparameter settings of the PPO algorithm in the Swimmer and Halfcheetah environments. The red stars indicate the normalized environment scores of a hyperparameter setting, which does well in Halfcheetah but poorly in Swimmer. The blue stars indicate the normalized scores of the hyperparameter setting, which maximizes the mean of the normalized environment scores across both environments.

malized environment score with hyperparameters tuned per environment. The *cross-environment tuned score* is the normalized environment score of the best fixed hyperparameter setting across the distribution of environments. We can use this notion of hyperparameter sensitivity to better understand new and existing algorithms. Reporting both hyperparameter sensitivity and the conventional performance-only evaluation metrics should provide a more complete picture of algorithm performance.

## 3.2 Sensitivity Analysis

Modern reinforcement learning algorithms are complex learning systems, and understanding them is a difficult task that requires multiple dimensions of analysis. Benchmark performance has been the primary metric (and often the only one) used for evaluating algorithms. However, this is only one dimension along which algorithms can be evaluated. Hyperparameter sensitivity is an important dimension to consider in the evaluation space, especially as practitioners begin to apply reinforcement learning algorithms to real-world applications. We propose the *performance-sensitivity plane* to aid in better understanding algorithms.

Consider the performance-sensitivity plane shown in Figure 4. To construct the plane, the center point is set to the hyperparameter sensitivity and per-environment tuned score of some reference point algorithm. We can consider how other algorithms relate to this reference point by considering which region of the plane they occupy. There are 5 regions of interest shaded by different colors and labeled numerically, which we will consider in turn.

An ideal algorithm would be both more performative and less sensitive. Therefore, algorithms that fall in **Region 1** (the top left quadrant) of the plane would be a strict improvement over the reference point algorithm. For some applications, perhaps additional sensitivity can be tolerated if the gains in performance are large enough. Algorithms that fall in **Region 2** are an example of this. The region represents algorithms whose increase in performance is greater than the corresponding increase in sensitivity. Conversely, for some applications sensitivity may matter a great deal and some performance loss can be endured. Algorithms that fall in **Region 3** are an example of those whose decrease in sensitivity outmatches their corresponding decrease in performance. Regions 1-3 represent algorithms that have notable redeeming qualities either in terms of performance, hyperparameter sensitivity, or both. However, perhaps a practitioner does not care about sensitivity. For example, they want to maximize the score of a specific benchmark, and hyperparameter tuning is no issue. Algorithms in **Region 4** may be adequate as they are algorithms that exhibit performance improvements and an even higher reliance upon per-environment hyperparameter tuning. Finally, those unfortunate algorithms that live in **Region 5** are in a space with both lower performance and higher sensitivity, making them undesirable.

A natural application of this diagram is to set the reference (center) point to the hyperparameter sensitivity and performance of some base algorithm and study how proposed modifications (or ablations) affect both sensitivity and performance. Often, new algorithms are created by modifying existing algorithms, such as normalizing targets, adding a regularization term to the loss function, gradient clipping, etc. We illustrate an example of this using PPO as a reference point in the next section.

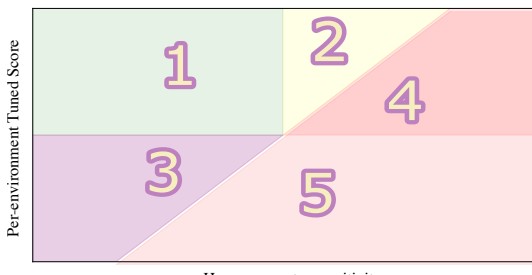

Figure 4: The performance-sensitivity plane for algorithmic evaluation. The center point indicates the hyperparameter sensitivity and performance of a reference point algorithm. The x-axis is the hyperparameter sensitivity metric as defined in equation 2. The y-axis is the per-environment tuned score (first term in equation 2). The diagonal line is the identity line shifted to intersect the reference point algorithm. The plane is then divided into 5 shaded regions that represent spaces of algorithms of varying qualities relative to the baseline.

## 4 Sensitivity Experiments

To illustrate the utility of the sensitivity analysis presented above, we performed an experiment to study the hyperparameter sensitivity and performance of several variants of the PPO algorithm, a widely used policy-gradient method in reinforcement learning (Schulman et al., 2017). We considered several normalization variants commonly used in PPO implementations (Andrychowicz et al., 2021; Huang et al., 2022) and some normalization variants, introduced in DreamerV3, that were purported to reduce hyperparameter sensitivity (Hafner et al., 2023; Sullivan et al., 2023).

### 4.1 Proximal Policy Optimization

PPO is an instance of an actor-critic method (Sutton & Barto, 2018) that maintains two neural networks: a policy network with parameters $\boldsymbol{\theta}$ and a value network with parameters $\mathbf{w}$. Given a policy $\pi_{\boldsymbol{\theta}_{old}}$, the agent performs a roll-out of $T$ steps, $(S_1, A_1, R_1, S_2, A_2, R_2, ...S_T, A_T, R_T)$, this rollout is then split into $m$ batches of length $k$. The critic is then updated via ADAM (Kingma & Ba, 2015) to optimize MSE loss with the truncated-$\lambda$ return as a target. The actor is updated via ADAM (Kingma & Ba, 2015) to optimize a clipped surrogate objective toward maximizing the expected return. An entropy regularizer is added to the actor loss to encourage exploration.

### 4.2 Normalization variants

Several normalization variants have been used in PPO implementations. We focus on three categories of normalization: observation normalization, value function normalization, and advantage normalization. An intuition behind value function or advantage normalization for hyperparameter sensitivity is that the scale and sparsity of rewards vary greatly across environments and that value function or advantage normalization should make actor-critic updates invariant to these factors possibly requiring less tuning of the step-size hyperparameters (van Hasselt et al., 2016). Another claimed benefit of advantage normalization variants is that by normalizing the advantage term, it is easier to find an appropriate value for the entropy regularizer coefficient $\tau$ across a distribution of environments (Hafner et al., 2023). Observation normalization standardizes the network inputs. This can mitigate large gradients, which may stabilize the learning system for hyperparameter tuning (Hafner et al., 2023; Andrychowicz et al., 2021), especially critic and actor step-size hyperparameters $\alpha_{\mathbf{w}}, \alpha_{\boldsymbol{\theta}} > 0$.

**Advantage per-minibatch zero-mean normalization**: A common implementation detail of PPO is per-minibatch advantage normalization (Huang et al., 2022). When performing an update, the advantage estimates used in the actor loss function are normalized by subtracting the mean of the advantage estimates in the sampled batch and dividing by the standard deviation of advantage estimates in the sampled batch.

**Advantage percentile scaling** : Another form of advantage normalization was introduced in the DreamerV3 (Hafner et al., 2023) ablations which divides the advantage estimate in the actor loss by a scaling factor. Exponential moving averages are maintained over the 95th and 5th percentiles of return estimates. The advantage term is divided by the difference of the two percentiles.

**Advantage lower bounded percentile scaling**: An alternate variant of percentile scaling is used in the DreamerV3 algorithm. Lower bounded percentile scaling applies a max operation to the percentile scaling factor, preventing the estimated advantage term from blowing up if the percentile difference falls below a threshold.

**Value target symlog**: DreamerV3 introduced a method of scaling down the magnitudes of target values by the symlog function. The symlog function and its inverse symexp are defined as:

$$\text{symlog}(x) \doteq \text{sign}(x)\ln(|x|+1) \qquad\qquad \text{symexp}(x) \doteq \text{sign}(x)(\exp(|x|-1) \tag{3}$$

As in DreamerV3, symlog is applied to the target in the critic loss and symexp is applied to the output of the critic network. In a subsequent study that applied DreamerV3 tricks to PPO (Sullivan et al., 2023), it was reported that the symlog transformation of the value target was one of the most impactful tricks in environments without reward clipping when applied to PPO.

**Observation zero-mean normalization**: A very common procedure with PPO is to normalize observations by maintaining running estimates of the mean and standard deviation of observations.

**Observation symlog**: DreamerV3 proposed an alternative form of observation normalization by applying the symlog function, compressing the observations.

A handful of prior work has investigated the benefits of similar algorithmic modifications to PPO. Previous work has reported the performance impact of the normalization variants commonly used with PPO: per-minibatch zero-mean advantage normalization and zero-mean observation normalization (Andrychowicz et al., 2021). Other work (Sullivan et al., 2023) investigated how PPO's performance is affected by the normalization variants introduced in DreamerV3 (lower bounded percentile scaling, value target symlog, and symlog observation) and that symlog was especially helpful in Atari when reward clipping is disabled. In addition, this work did not perform any hyperparameter tuning for the variants of PPO they tested. In our results, hyperparameter tuning demonstrated a significant effect on the relative performance of these algorithms.

Given these normalization variants, a natural question that arises is how do they affect the hyperparameter sensitivity of PPO? To the best of our knowledge, a careful study of the effect these normalization variants have on the hyperparameter sensitivity of PPO has yet to be done and it provides a good test of our new methodology.

### 4.3 Sensitivity Experiment with PPO variants

We investigated the effect of each of the described normalization variants on PPO. To isolate these effects, we did not apply reward clipping, reward scaling, or observation normalization wrappers by default. We focused our attention on four critical hyperparameters of PPO: the step-size for the critic $\alpha_{\mathbf{w}}$, the step-size for the actor $\alpha_{\boldsymbol{\theta}}$, the coefficient of the entropy regularizer $\tau$, and the truncated-$\lambda$ return mixing parameter $\lambda$. We performed a large grid search spanning five orders of magnitude across five Brax Mujoco domains. Near the extreme endpoints of the grid search, some hyperparameter configurations diverged. We ignored hyperparameter combinations that caused a particular algorithm to diverge over 10% of the time. We averaged the performance over the non-diverging runs.

Consider the performance-sensitivity plane in Figure 5. The reference point at the center is the hyperparameter sensitivity and performance found for PPO without normalization. The error bars displayed indicate 95% confidence intervals formed from a 10,000 sample bootstrap. First, note that none of the normalization variants resulted in an improvement that both raised performance and lowered sensitivity. All forms of advantage normalization increased performance. However, this performance gain comes with a trade-off: increased hyperparameter sensitivity. The marginal gain in performance per unit of increased sensitivity varied between advantage normalization methods. Advantage per-minibatch zero-mean normalization had a greater increase in performance than sensitivity (Region 2). Both percentile scaling-based variants of advantage normalization resulted in more significant sensitivity increases than performance increases (Region 4), indicating an enhanced reliance on hyperparameter optimization methods. Applying the symlog function to the value target lowered performance and may have slightly increased sensitivity (Region 5). It may, however, be the case that the choice of environment distribution did not have enough variation in reward magnitude for the utility of value target symlog to be demonstrated. It appears that observation normalization may slightly reduce sensitivity, although it is unclear due to the width of the confidence intervals.

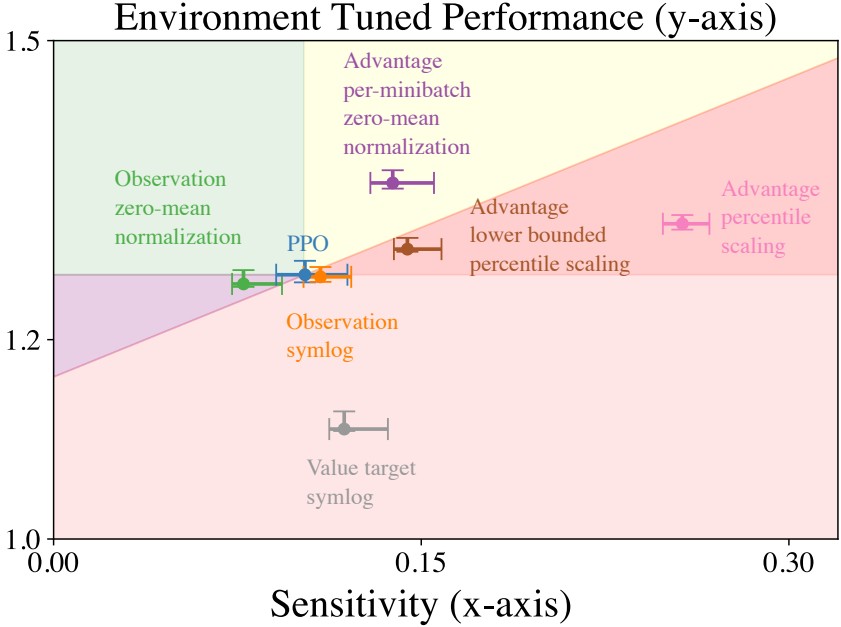

Figure 5: Performance-sensitivity plane with unnormalized PPO as the center reference point. Variants of PPO plotted. The x-axis indicates hyperparameter sensitivity as defined in equation 2. The y-axis represents the per-environment tuned score (first term in the sensitivity calculation of equation 2). Hyperparameter sensitivity and per-environment tuned score metrics were computed from a 200 run sweep of 625 hyperparameter settings across 5 Brax Mujoco environments (Ant, Halfcheetah, Hopper, Swimmer, and Walker2d). Error bars show the endpoints of 10,000 sample 95% bootstrap confidence intervals around both the performance and hyperparameter sensitivity metrics (two dimensions).

The performance-sensitivity plane provides insights into how algorithmic changes may alter an algorithm's reliance on per-environment hyperparameter tuning with no additional computational expense other than what is already required from a standard hyperparameter tuning procedure.

The sensitivity metric is intimately tied to the chosen environment distribution, and our findings are thus limited to the Brax Mujoco environments tested. We argue the restricted environment distribution is not just a practical choice, but was a feature of the study. Somewhat surprisingly we found significant sensitivity across a distribution of fairly similar environments. In Figure 7 displayed in Appendix D, we perform a leave-one-out study reporting how the performance-sensitivity plane changes when each of the five environments is dropped from the data. Observe that while the exact values of the points shift, their position relative to the reference point remains mostly unchanged.

The performance-sensitivity plane allows for a richer understanding of algorithms than performance-only evaluation procedures, but it does not capture the full picture. Two algorithms can sit in the same location on the plane and yet have very different hyperparameter characteristics. Consider the case where there are two algorithms. The first algorithm is highly sensitive with respect to one hyperparameter, which needs to be carefully tuned per environment. The second algorithm has the same sensitivity but needs to be tuned per environment for dozens of hyperparameters with complex interactions. The hyperparameter sensitivity would not differentiate between these two algorithms. An additional metric is needed.

## 5 Effective Hyperparameter Dimensionality

There are many cases where a practitioner can tune some but not all of an algorithm's tunable hyperparameters. It may be the case that if a few key hyperparameters are tuned per environment, then a preponderance of an algorithm's potential performance can be gained. This motivates the definition of *effective hyperparameter dimensionality*, a metric that measures how many hyperparameters must be tuned in order to obtain near-peak performance.

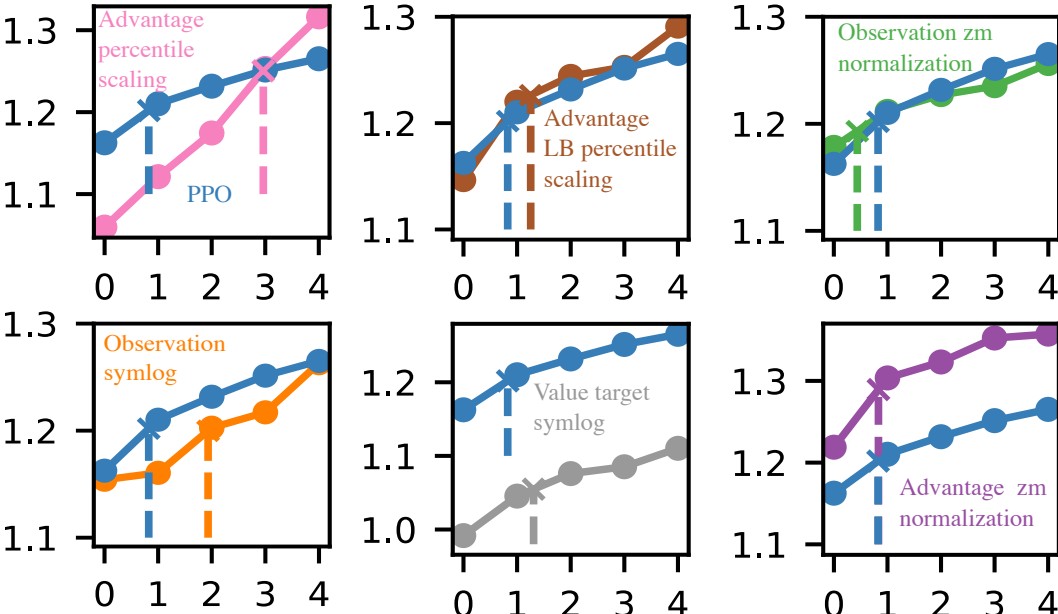

Figure 6: Normalized performance scores as a function of the number of hyperparameters tuned per environment. The subplots compare PPO to the PPO variants studied. The x-axis indicates the size of the subset of hyperparameters being tuned. The y-axis is the average normalized score across the environment distribution. Each dot indicates the normalized score obtained by tuning the most performant subset of hyperparameters of each size. The curve is an interpolation between the dots. The dashed line indicates the point at which the curve reaches 95% of peak performance. LB is an abbreviation for lower bounded, zm is an abbreviation for zero-mean.

For a given algorithm $\omega$ with hyperparameter space $H^\omega$, number of tunable hyperparameters $n(\omega)$, and environment distribution $\mathcal{E}$, let $h^* \doteq \arg\max_{h \in H^\omega} \frac{1}{|\mathcal{E}|} \sum_{e \in \mathcal{E}} \Gamma(\omega, e, h)$ be the hyperparameter setting which maximizes the cross-environment tuned normalized score. Effective hyperparameter dimensionality $d(\omega)$ is defined as

$$d(\omega) = n(\omega) - \max_{\{h^e \in H^\omega\}_{e \in \mathcal{E}}} \sum_{i=1}^{n(\omega)} \mathbb{1}[h_i^e = h_i^* \ \forall e \in \mathcal{E}]$$

$$\text{s.t.} \quad \frac{1}{|\mathcal{E}|} \sum_{e \in \mathcal{E}} \Gamma(\omega, e, h^e) \geq 0.95 \frac{1}{|\mathcal{E}|} \sum_{e \in \mathcal{E}} \max_{h \in H^\omega} \Gamma(\omega, e, h)$$

(4)

The effective hyperparameter dimensionality of an algorithm measures the minimal number of hyperparameters that must be tuned per-environment while retaining the majority of the performance that can be realized by tuning all hyperparameters per environment. The threshold of 95% peak performance can be changed at a practitioner's discretion to whatever meets their performance requirements. To compute effective hyperparameter dimensionality, one needs to consider subsets of hyperparameters to find a minimum subset that achieves the required performance threshold.

For the same algorithmic variants of PPO as studied above, Figure 6 displays normalized scores as a function of the number of hyperparameters tuned per environment, choosing the most performant subset to tune. Table 2 in Appendix C provides a listing of the most performant subsets of varying sizes observed during this experiment. The curve interpolates between the normalized scores. The vertical dashed line indicates the point along the curve that reaches 95% of the per-environment tuned score. In the case of advantage percentile scaling, modifying PPO with the normalization variant moves the point to the right, indicating this variant improves performance at the cost of increasing pressure on the number of hyperparameters necessary to tune. Also, note how the performance

ranking can shift based on the number of hyperparameters that have been tuned; such as with PPO and the advantage percentile scaling variant (top left plot). For some variants, performance flattens after tuning only three hyperparameters. Whereas for other variants, performance is almost linear in the number of hyperparameters tuned, suggesting sensitivity to all hyperparameters. On the performance-sensitivity plane (Figure 5 in the previous section), both the advantage percentile scaling and advantage lower-bounded percentile scaling variants fall in the same region (Region 4). Yet, in Figure 6, we can see that the advantage lower bounded percentile scaling variant can obtain higher performance levels than the advantage percentile scaling variant when tuned on smaller subsets of hyperparameters. This observation that algorithms can have similar hyperparameter sensitivities and vastly different effective hyperparameter dimensionalities indicates the power of using both metrics for studying algorithms.

## 6    Limitations and Future Work

The hyperparameter sensitivity and effective hyperparameter dimensionality metrics will depend heavily on several important empirical design choices. The premise of both metrics is that a practitioner is concerned with understanding sensitivity with respect to a distribution of environments that they care about. If the distribution of environments changes, the metrics will need to be evaluated with respect to the new distribution. This dependence on the environment distribution could be exploited, as someone could artificially make an algorithm appear less sensitive by including several easy environments that all hyperparameter settings will do well in, although score normalization will somewhat counteract this. In addition, the level of granularity with which hyperparameter sweeps are performed will have an effect on both metrics. Another factor that can impact the metrics is the choice of the score normalization method used. A practitioner could use other score normalization methods and the resulting sensitivity scores may be different.

A next step is to apply the proposed sensitivity and dimensionality metrics to a larger set of algorithms and environments. Related to this work is the literature of AutoRL (Eimer et al., 2023). The goal of AutoRL is to tune hyperparameters via some hyperparameter optimization algorithm (which make use of their own hyperparameters—hyper-hyperparameters). Future work could use the definitions provided here to try to understand if the algorithms proposed in the AutoRL literature reduce sensitivity over the base algorithms that are being modified—measuing the sensitivity of the hyper-hyperparameters. A study comparing the sensitivities and dimensionalities of AutoRL methods to the sensitivities and dimensionalities of the base learning algorithms they optimize would be prudent.

## 7    Conclusion

As learning systems become more complicated, careful empirical practice is critical. Modern reinforcment learning algorithms contain numerous hyperparameters whose interactions and sensitivities are not well understood. Common practice, which is focused on achieving state-of-the-art performance, risks overfitting to benchmark tasks and overly relying on hyperparameter optimization. Most empirical work in reinforcement learning has focused only on evaluating algorithms based on benchmark performance, leaving the effects of hyperparameters under-studied. In this work, we propose a new evaluation methodology based on two metrics that allow practitioners to better understand how an algorithm's performance relates to its hyperparameters. We show how this methodology is useful in evaluating methods purported to mitigate sensitivity. We identify that the studied advantage normalization methods, while improving performance, also increase hyperparameter sensitivity and can increase the number of sensitive hyperparameters.

## 8    Acknowledgments

The authors would like to thank Ana Paola Garcia Alonzo for assistance in formatting figures. The authors would like to thank David Sychrovský and Anna Hakhverdyan for useful discussions. This research was supported by grants from the Alberta Machine Intelligence Institute (Amii); Canada CIFAR AI Chairs, Amii; and NSERC. Compute was made available by the Digital Research Alliance of Canada.

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

# A   Broader Impact Statement

Hyperparameter sweeps and per-environment tuning are the most computationally expensive and environmentally impactful parts of reinforcement learning research. Our study ran for approximately 4.5 GPU years on NVIDIA 32GB V100s. While this is substantial, we believe that using compute to better understand the sensitivity of current algorithms is an essential step towards developing more environmentally friendly algorithms. This work investigated an empirical methodology for evaluating the hyperparameter sensitivity of reinforcement learning agents. The immediate societal impact is minimal. However, our methodology may aid in developing performative algorithms with low hyperparameter sensitivity. If this occurs, these algorithms will result in less need for hyperparameter tuning and, as a result, have a positive impact on lowering the carbon footprint of reinforcement learning experiments.

# B   Proliferation of Hyperparameters

There is a trend in which the current state-of-the-art algorithms often contain more hyperparameters than the previous state-of-the-art. Table 1 lists hyperparameter counts for representative algorithms from each of the three main categories of reinforcement learning methods: values-based, policy-gradient, and model-based.

# C   Hyperparameter Sweep Details

The PPO implementation used was heavily inspired by the PureJaxRL PPO implementation (Lu et al., 2022). The variants advantage per-minibatch zero-mean normalization and observation zero-mean normalization are the standard implementations provided within PureJaxRL. The variants: symlog observation, symlog value target, percentile scaling, and lower bounded percentile scaling closely follow the implementation of the DreamerV3 tricks applied to PPO shown in Sullivan et al. (2023) as well as referencing the original DreamerV3 repository Hafner et al. (2023).

The policy and critic networks were parametrized by fully connected MLP networks, each with two hidden layers of 256 units. The network used the tanh activation function. Separate ADAM optimizers (Kingma & Ba, 2015) were used for training the actor and critic networks. The environments used in the experiments were the Brax implementations of Ant, Halfcheetah, Hopper, Swimmer, and Walker2d.(Freeman et al., 2021). The hyperparameter sweeps were grid searches over eligibility trace $\lambda \in \{0.1, 0.3, 0.5, 0.7, 0.9\}$, entropy regularizer coefficient $\tau \in \{0.001, 0.01, 0.1, 1.0, 10.0\}$, actor step-size $\alpha_{\boldsymbol{\theta}} \in \{0.00001, 0.0001, 0.001, 0.01, 0.1\}$, and critic step-size $\alpha_{\mathbf{w}} \in \{0.00001, 0.0001, 0.001, 0.01, 0.1\}$. Each run lasted for 3M environment steps. 200 runs were performed for each of the algorithms, environments, and hyperparameter settings. Like the PureJaxRL PPO implementation, the entire training loop was implemented to run on GPU. We will release code and experiment data at `https://github.com/jadkins99/hyperparameter_sensitivity`, promoting the further investigation of hyperparameter sensitivity in the field of reinforcement learning.

| Algorithm name | Year | Number of hyperparameters | Comments |
|---|---|---|---|
| Q-learning | 1989 | 3 | (+1) step size, (+1) $\epsilon$-greedy exploration, (+1) discount factor (Watkins, 1989) |
| R-learning | 1993 | 3 | (+2) step sizes, (+1) $\epsilon$-greedy exploration (Schwartz, 1993) |
| LSPI | 2003 | 4 | (+1) discount factor (+1) stopping condition, (+1) $\epsilon$-greedy exploration (+1) initialization parameter (Lagoudakis & Parr, 2003) |
| NFQ | 2006 | 5 | (+1) discount factor (+1) $\epsilon$-greedy exploration, (+2) Uses RProp, (+1) number of inner-loop iterations (Riedmiller, 2005) |
| DQN | 2013 | 16 | Hyperparameter table from paper. |
| PER | 2016 | 20 | (+16) inherited from DQN, (+4) added for PER. |
| Rainbow | 2017 | 25 | (+20) inherited from PER, (+5) (1 extra from changing optimizer from RMSProp to ADAM, n step return, 3 from distributional RL). |
| Agent57 | 2020 | 34 | Hyperparameter table from paper. |
| Actor-critic | 1983 | 3 | (+1) actor step size, (+1) critic step size, (+1) discount factor. |
| Actor-critic with eligibility trace | 1983 | 5 | (+1) actor step size, (+1) critic step size, (+1) discount factor, (+2) eligibility traces . |
| A3C | 2016 | 11 | (+3) Uses RMSProp, (+1) value function loss coefficient, (+1) entropy coefficient, (+1) number of actors, (+1) batch size, (+1) gradient clipping, (+1) atari frame stacking, (+1) target network update rate, (+1) discount factor . |
| PPO-clip | 2017 | 24 | Hyperparameter table from paper. Additional hyperparameters discussed in ICLR blogpost. (+11) inherited from A3C, (+1) from switching to ADAM, (+1) $\lambda$ returns, (+1) number of epochs, (+1) actor loss clipping, (+1) value loss clipping, (+2) Log stdev of action distribution LinearAnneal(-0.7,-1.6), (+1) ADAM LR annealing, (+1) reward scaling, (+2) reward clip, (+2) post-normalization observation clip. |
| SAC | 2018 | 22 | Stable Baselines documentation. (+10) inherited from A3C, (+1) ADAM, (+1) number of epochs, (+1) reward scaling, (+2) reward clip, (+2) post-normalization observation clip, (+1) entropy coefficient step-size, (+1) target entropy, (+2) action Gaussian noise, (+1) polyak update. |
| Tabular Dyna | 1991 | 4 | (+1) $\epsilon$-greedy, (+1) discount factor, (+1) number of planning steps, (1) stepsize |
| Tabular Dyna+ | 1991 | 5 | (+1) $\epsilon$-greedy, (+1) discount factor, (+1) number of planning steps, (+1) stepsize (+1) planning exploration reward bonus |
| Linear Dyna | 2012 | 6 | (+1) $\epsilon$-greedy, (+1) discount factor, (+1) number of planning steps, (+3) stepsizes (papers uses one stepsize but there it is used for transition matrix, expected reward vector, and value function estimate |
| DreamerV1 | 2020 | 26 | Paper. (+1) number random seed episodes, (+1) number of update steps, (+1) buffer size, (+1) batch size, (+1) batch length, (+1) imagination horizon, (+1) $\lambda$, (+1) actor step-size, (+1) critic step-size, (+3) ADAM, (+1) RSSM units, (+1) discount factor, (+1) entropy coefficient, (+1) target network update, (+1) action model tanh scale factor, (+1) step-size for world model, (+1) gradient clipping, (+1) KL regularizer clipping, (+2) action Gaussian noise, (+1) action repeat, (+3) discrete action $\epsilon$-greedy linear decay over first 200k steps. |
| DreamerV2 | 2022 | 28 | Modification summary. (+26) inherited from DreamerV1, (+1) number of discrete latents, (+1) KL balancing, (+1) actor gradient mixing, (+1) weight decay, (-2) removed action Gaussian noise. |
| DreamerV3 | 2023 | 41 | Hyperparameter table from paper. Modification summary. (+28) inherited from DreamerV2, (+1) World model loss clipping, (+1) actor unimix random exploration, (+3) advantage percentile scaling (decay rate, lower bound,(p,1-p) percentile scaling), (+2) two additional $\beta$ terms in world model loss, (+1) Critic EMA decay, (+1) Critic EMA regularizer coefficient, (+2) Adaptive Gradient Clipping, (+1) Critic replay loss scale, (+1) Critic loss scale, (+1) Actor loss scale, (+1) latent unimix, (-2) same step-size for actor, critic, and world model. |

Table 1: This table reports counts of hyperparameters (excluding those relating to neural network architecture) from a sampling of prominent algorithms proposed over the last decade. When possible, we tried to use hyperparameter tables listed in the original papers. Otherwise, we used documentation from popular implementations. The comments column contains links to sources used.

| Algorithm Variant | Size 1 Subsets | Size 2 Subsets | Size 3 Subsets |
|---|---|---|---|
| PPO | $\lambda$ | $\tau, \lambda$ | $\tau, \lambda, \alpha_{\mathbf{w}}$ |
| Observation symlog | $\lambda$ | $\tau, \lambda$ | $\tau, \lambda, \alpha_{\mathbf{w}}$ |
| Observation zero-mean normalization | $\lambda$ | $\tau, \lambda$ | $\tau, \lambda, \alpha_{\boldsymbol{\theta}}$ |
| Advantage percentile scaling | $\alpha_{\mathbf{w}}$ | $\lambda, \alpha_{\mathbf{w}}$ | $\lambda, \alpha_{\boldsymbol{\theta}}, \alpha_{\mathbf{w}}$ |
| Advantage lower bounded percentile scaling | $\alpha_{\mathbf{w}}$ | $\lambda, \alpha_{\mathbf{w}}$ | $\lambda, \alpha_{\boldsymbol{\theta}}, \alpha_{\mathbf{w}}$ |
| Advantage per-minibatch zero-mean normalization | $\lambda$ | $\lambda, \alpha_{\mathbf{w}}$ | $\tau, \lambda, \alpha_{\mathbf{w}}$ |
| Value target symlog | $\alpha_{\boldsymbol{\theta}}$ | $\tau, \lambda$ | $\tau, \lambda, \alpha_{\mathbf{w}}$ |

Table 2: The subsets of hyperparameters that were found to be most impactful to tune per environment as measured when creating Figure 6.

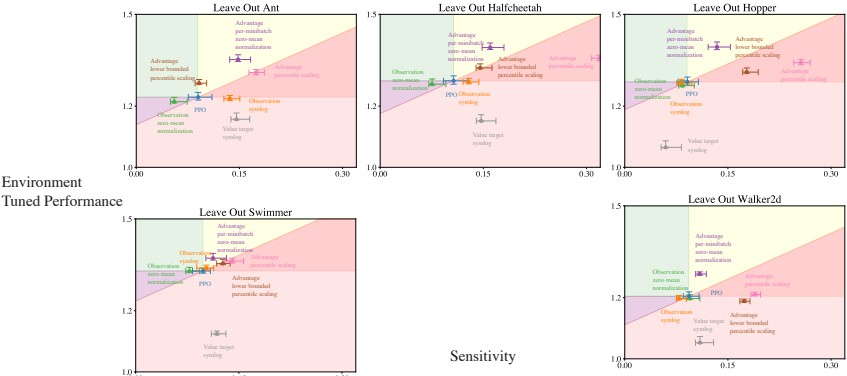

Figure 7: Performance-sensitivity planes shown are formed by leaving out each of the five environments. Error bars are 95% confidence intervals obtained from 1000 sample bootstraps.

# D    Additional Figures

The sensitivity metric is ultimately tied to a distribution of environments, and our findings are thus limited to Brax Mujoco environments tested. Claims cannot be about environments outside the evaluated environment set. One may wonder about the stability of the results if the environment distribution changes on a small scale. In Figure 7, we repeat the sensitivity metric for each of the five environment subsets that can be obtained by dropping a single environment. Leaving out environments does shift the reference point, and the position of the variants shifts somewhat relative to the reference point. However, the regional position of the variants relative to the reference point is mostly consistent.

In Figure 8, we also include results with final performance, i.e., the sum of rewards obtained over the final 1000 timesteps of learning (the truncation length). The performance of reinforcement learning agents does not monotonically increase. In many cases, with specific hyperparameters, it may collapse in the middle or end of learning. Therefore, this metric is noisy, and it is difficult to reason about where algorithms lie according to it on the performance sensitivity curve.

| Algorithm Variant | Environment | $\lambda$ | $\tau$ | $\alpha_{\boldsymbol{\theta}}$ | $\alpha_{\mathbf{w}}$ | Mean of Returns | Standard Deviation of Returns |
|---|---|---|---|---|---|---|---|
| Advantage percentile scaling | ant | 0.9 | $10^{-3}$ | $10^{-4}$ | $10^{-4}$ | 34.64 | 8.54 |
| Advantage percentile scaling | halfcheetah | 0.3 | $10^{-3}$ | $10^{-4}$ | $10^{-3}$ | 2220.81 | 179.74 |
| Advantage percentile scaling | hopper | 0.9 | $10^{-3}$ | $10^{-5}$ | $10^{-3}$ | 1005.48 | 41.22 |
| Advantage percentile scaling | swimmer | 0.9 | $10^{-2}$ | $10^{-2}$ | $10^{-3}$ | 35.34 | 5.83 |
| Advantage percentile scaling | walker2d | 0.9 | $10^{-2}$ | $10^{-4}$ | $10^{-3}$ | 846.72 | 94.99 |
| Advantage lower bounded percentile scaling | ant | 0.7 | $10^{-3}$ | $10^{-4}$ | $10^{-4}$ | 35.53 | 11.22 |
| Advantage lower bounded percentile scaling | halfcheetah | 0.3 | $10^{-3}$ | $10^{-4}$ | $10^{-3}$ | 2216.12 | 176.56 |
| Advantage lower bounded percentile scaling | hopper | 0.9 | $10^{-2}$ | $10^{-5}$ | $10^{-3}$ | 1003.62 | 46.69 |
| Advantage lower bounded percentile scaling | swimmer | 0.9 | $10^{-3}$ | $10^{-2}$ | $10^{-4}$ | 41.24 | 23.59 |
| Advantage lower bounded percentile scaling | walker2d | 0.9 | $10^{-3}$ | $10^{-4}$ | $10^{-3}$ | 831.81 | 140.81 |
| Advantage per-minibatch zero-mean normalization | ant | 0.7 | $10^{-2}$ | $10^{-4}$ | $10^{-4}$ | 36.98 | 10.37 |
| Advantage per-minibatch zero-mean normalization | halfcheetah | 0.3 | $10^{-3}$ | $10^{-4}$ | $10^{-3}$ | 2369.19 | 169.13 |
| Advantage per-minibatch zero-mean normalization | hopper | 0.9 | $10^{-3}$ | $10^{-5}$ | $10^{-3}$ | 1006.39 | 30.88 |
| Advantage per-minibatch zero-mean normalization | swimmer | 0.5 | $10^{-2}$ | $10^{-4}$ | $10^{-5}$ | 41.11 | 3.29 |
| Advantage per-minibatch zero-mean normalization | walker2d | 0.7 | $10^{-2}$ | $10^{-4}$ | $10^{-3}$ | 824.87 | 96.69 |
| PPO | ant | 0.7 | $10^{-2}$ | $10^{-4}$ | $10^{-3}$ | 38.95 | 9.94 |
| PPO | halfcheetah | 0.5 | $10^{-2}$ | $10^{-4}$ | $10^{-3}$ | 2311.89 | 172.22 |
| PPO | hopper | 0.9 | $10^{-3}$ | $10^{-5}$ | $10^{-3}$ | 1002.97 | 71.25 |
| PPO | swimmer | 0.9 | $10^{-3}$ | $10^{-4}$ | $10^{-4}$ | 33.41 | 9.41 |
| PPO | walker2d | 0.9 | $10^{-1}$ | $10^{-4}$ | $10^{-3}$ | 761.36 | 92.23 |
| Observation zero-mean normalization | ant | 0.7 | $10^{-2}$ | $10^{-4}$ | $10^{-3}$ | 39.78 | 9.85 |
| Observation zero-mean normalization | halfcheetah | 0.5 | $10^{-2}$ | $10^{-4}$ | $10^{-3}$ | 2306.22 | 161.09 |
| Observation zero-mean normalization | hopper | 0.9 | $10^{-3}$ | $10^{-5}$ | $10^{-3}$ | 1004.47 | 68.99 |
| Observation zero-mean normalization | swimmer | 0.9 | $10^{-2}$ | $10^{-2}$ | $10^{-4}$ | 33.63 | 2.14 |
| Observation zero-mean normalization | walker2d | 0.9 | $10^{-1}$ | $10^{-4}$ | $10^{-3}$ | 755.25 | 88.62 |
| Value target symlog | ant | $10^{-1}$ | $10^{-3}$ | $10^{-4}$ | $10^{-4}$ | 8.57 | 2.48 |
| Value target symlog | halfcheetah | 0.7 | $10^{-2}$ | $10^{-4}$ | $10^{-4}$ | 1731.16 | 149.42 |
| Value target symlog | hopper | 0.9 | $10^{-1}$ | $10^{-5}$ | $10^{-3}$ | 1067.40 | 57.11 |
| Value target symlog | swimmer | 0.5 | $10^{-3}$ | $10^{-4}$ | $10^{-5}$ | 34.67 | 1.00 |
| Value target symlog | walker2d | 0.9 | $10^{-1}$ | $10^{-4}$ | $10^{-3}$ | 698.09 | 85.97 |
| Observation symlog | ant | 0.7 | $10^{-2}$ | $10^{-4}$ | $10^{-3}$ | 39.57 | 11.15 |
| Observation symlog | halfcheetah | 0.5 | $10^{-2}$ | $10^{-4}$ | $10^{-3}$ | 2321.00 | 147.05 |
| Observation symlog | hopper | 0.9 | $10^{-2}$ | $10^{-5}$ | $10^{-3}$ | 1007.13 | 69.63 |
| Observation symlog | swimmer | 0.9 | $10^{-3}$ | $10^{-4}$ | $10^{-4}$ | 32.90 | 9.60 |
| Observation symlog | walker2d | 0.9 | $10^{-1}$ | $10^{-4}$ | $10^{-3}$ | 767.30 | 77.60 |

Table 3: This table reports means and standard deviations across seeds of the average return observed during learning for a subset of the different hyperparameter settings, algorithm variants, and environments.

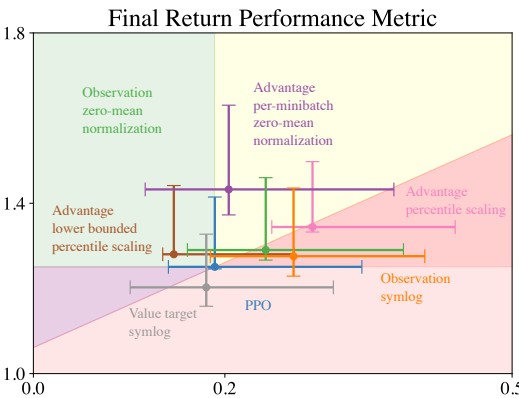

Figure 8: Performance-sensitivity plane with final return as the performance metric. Variants of PPO plotted. The x-axis indicates hyperparameter sensitivity as defined in equation 2. The y-axis represents the per-environment tuned score (first term in the sensitivity calculation of equation 2). Error bars are 95% confidence intervals from a 1000 sample bootstrap.

