# OpenReview forum: "A Method for Evaluating Hyperparameter Sensitivity in Reinforcement Learning"
_NeurIPS.cc/2024/Conference — NeurIPS 2024 poster_

### Official Review · Reviewer_ydma · 2024-06-16

**Soundness:** 2
**Presentation:** 3
**Contribution:** 3
**Rating:** 4
**Confidence:** 4

**Summary:**

The authors define a environment-parameter sensitivity metric in terms of a Jensen gap of min-max scaled performance. The gap gives us the difference between the best average performance and the average best performance over a particular algorithms hyperparameters over a set of environments. The authors then provide a 2D visualization where they plot this sensitivity gap against the average best-performance. This gives an intuitive way to classify whether a change in parametersensitivity can be outweighed by its (positive) change in performance.

The authors run experiments for PPO on brax to illustrate their method.

At the end, the authors also use their performance metric to tune subsets of the total parameter set.

**Strengths:**

The definition for the environmnent-dependent parameter sensitivity is intuitive through its use of Jensen's gap. I agree that this is a sound method to evaluate robustness of parameter-tuning. Although, care must be taken when choosing the environment-sets.

Figure 3 and the discussion in 3.2 is excellently presented, and gives an intuitive way to think about what patterns we would like to see when performing parameter ablations.

The experimental setup is repeated over many repetitions and use appropriate statistical methods, when discussing the results, the authors discuss overlapping intervals and are decently careful about making conclusions.

**Weaknesses:**

### Summary
Although this paper shows a nice way to visualize robustness of parameter optimization methods across sets of environments, the severe computation requirements needed to create these visualizations on top of the shallow experimental setup that the authors performed, makes it unlikely that the current results will prove useful to other practitioners (aside from the isolated case applying PPO to Brax).

The chosen metric for performance (AUC) is incomplete when considering how RL is often applied (when only caring about final performance), and arguably estimated using a non-robust bound estimator. Furthermore, the authors only test on 5 Brax mujoco environments when comparing extensions that originate from Atari, DMLab, Minecraft or elsewhere. The authors, do not take characteristics of the environments they test on into account when discussing results. As a consequence, the conclusions we can take from the presented results are quite limited.

Therefore I vote for reject.

### Major comments
 - The single choice of AUC metric is poorly motivated, although it is a relevant metric, in RL we are also often only interested in **final performance**. So, the AUC gives a proxy for cumulative regret, I also want to see a proxy for simple regret, the best-run within a algorithm's training cycle.
 - The authors argue that we need robust estimators of performance for hyperparameter sensitivity. So, I am worried about the choice for min-max normalization of the algorithms' performance, see Eq. 1). These can be extremely high-variance estimators for the performance bounds, and therefore, aren't really robust (perhaps explaining why the authors needed 200 seeds per ablation)... I understand that this always nicely scales results into $[0, 1]$, however, if we rerun the experiments (different seeds and parameter-sets), would Figures 1 and 2 give similar patterns? Would this change our conclusion about what choice of hyperparameter is better in terms of environment robustness? Why not choose for an interquantile range or even a $[p, 1-p]$ percentile range that has guaranteed lower asymptotic variance?
 - The authors don't discuss the sensitivity/ robustness to the choice of environment sets for Eq.2). I.e., what characteristics do the environments have that we test robustness for. For this reason, I also think that the discussion in 4.3 is a bit shallow considering the choice of Mujoco environments. The rewards in Brax are relatively well scaled and behaved (at least compared to e.g., a game like 2048 which has exponential rewards), so concluding that the value target symlog is not beneficial (on top of my comment about min-max scaling) should be nuanced and discussed appropriately.
 - It's not clear from Eq.5) (and the whole of section 5) how we decide what reference point to take when deciding which parameters to tune. I.e., when we choose to keep some parameters fixed, what should their values be? Are the defaults by definition robust or not?


### Technical comments
 - Definition of normalized environment score $\Gamma$ is wrong, why is the action-space $A$ used here? (see section 2). Why is the symbol for environment sets not consistent, why use $E$ instead of $\mathcal{E}$. In section 5 suddenly $\Omega$ is used again in place of $A$.


### Minor comments
 - Eq. 2) Why not write the sum as an expectation, $\mathbb{E}[\max_h \Gamma(...) | e \sim {p(e)}] - \max_h \mathbb{E}_{p(e)} \Gamma(...)$. I.e., the more recognizable Jensen gap.
 - Shortly discuss expected behaviour of Eq.2 and what it means, i.e., a value of 1 means a large gap, meaning not robust, and a value of 0 means no gap, meaning very robust.
 - Line 220, minibatch advantage normalization is not that important for PPO, even the blog-post that the authors cite here says so.
 - What ranges for the parameters were chosen in section 4.1 ? It's easy to break an algorithm through a bad or uninformed choice for the learning rates... It's also not really clear from the text how the authors created the algorithm configurations to test, i.e., random-search? an exhaustive grid? (Found this in Appendix D, the ranges for the values are OK when speaking from experience, and the authors used exhaustive grid-search; this **must** be included in the main text)
 - Line 291, "It may ... be gained." This sentence is broken, maybe rephrase: "By finetuning a few important parameters for each specific environment, you can unlock most of an algorithm's best performance."

**Questions:**

- Could the authors change the bound calculation (e.g., IQR or something similar), this should be an easy local change, and add the best-performing run metric (i.e., have the AUC and the best-run per algorithm side-by-side).
- Could the authors improve on the discussion and conclusions of the results, discussing the characteristics of the environment and how the choice of parameters hypothetically covaries with this.

**Limitations:**

Section 6 shortly discuss limitations, however, the actual limitations are discussed in appendix D. The shallow experimental setup that was performed resulted in 7 GPU years on modern NVIDIA GPUs. Since most code was implemented in Jax and the algorithm used was PPO, one of the most light-weight RL algorithms currently out there, the current framework does not offer much headroom for testing other methods.

The severe computation costs are not included in the broader impact statement. The authors do state that reducing carbon footprint of DRL experiments are important, however, the presented methodology is not carbon friendly. If, the authors presented a better experimental setup, then this would alleviate the need for other practitioners from running these types of sensitivity analyses for PPO. However, this is currently not the case. Another way to improve the current setup is to use variance reduction methods in some way to reduce the immense compute requirements.

---

> ### Author Rebuttal · Authors · 2024-08-07
>
> Thank you very much for your review and comments on improving our work.
>
>
> ## Major comments:
>
> 1. Several performance metrics are common in reinforcement learning: best policy performance, performance during final k episodes, AUC, etc. We chose AUC because it informs us about the rate of learning. For example, in continual RL, the goal is not the final policy—there is none—and the main interest is learning, not the outcome of learning. Many applications of RL are similar, for example, in data-center cooling, learning happens in deployment. Another empiricist could care about final policy performance and measure sensitivity to that. We can add sensitivity analysis that uses final performance as a metric to the appendix for the final version.
>
> 2. This is a valid question regarding the method's stability regarding the environment set and score normalization. We used the statistical bootstrap specifically to represent the variance in min-max estimation.  But, normalizing by a (p,1-p) percentile range may still be a better choice; we will add results with a (p,1-p) normalization. Thank you for the suggestion.
>
> 3. Different environment sets may produce different results. Min-max normalization, in particular, may be inappropriate in environments with exponential rewards scaling. In such cases, another choice of normalization may be preferable.
> Regarding value target symlog’s benefit, our results and conclusions are indeed tied to the Mujoco environments we chose. We tried to avoid making general conclusions in the paper about the algorithms beyond the empirical setting that we tested in. If there is a sentence that is particularly problematic concerning this, please let us know. This is missing the important point that even in one common environment suite, sensitivity emerges and is a big deal. We chose PPO and Brax Mujoco to demonstrate that even with one of the most heavily studied algorithms in one of the most used environment suites, these definitions can provide insight.
> As for “What characteristics do the environments have that we test robustness for” is a massive question—a paper all on its own—and we would argue that RL researchers have made little progress on this important but separate question.
>
> 4. In section 5, When leaving hyperparameters fixed, we set them to the values that maximize cross-environment performance (i.e. the default). The intuition behind the definition of effective hyperparameter dimensionality (and thus all of section 5) is trying to measure how many hyperparameters are “robust” and able to be left to default. Indeed, another huge challenge in RL: how do we establish sensible defaults.
>
> ## Technical comment
> 1. You are correct; this is wrong. This typo resulted from a change of notation for the algorithm set. This will be fixed for the camera-ready version.
>
> ## Minor Comments
> 1.  Thank you for this suggestion. The notation change and noting the definition as a Jensen gap would be a nice improvement. We will change for the camera-ready version.
> 2. We can add a sentence or two of extra interpretation of the ranges of the sensitivity gap in the discussion. Thank you for the suggestion.
> 3. The blog post states that minibatch advantage normalization does not have much effect on performance. To the best of our knowledge, its relevance for sensitivity has not been well-studied. However, some intuition suggests that it may be important for making the entropy coefficient less sensitive. This was a stated [motivation](https://arxiv.org/pdf/2301.04104#page=6) for the (albeit different) form of advantage normalization performed in DreamerV3.
> 4. Thank you for letting us know this was a point of confusion. We will add to the main text that a grid search was performed.
> 5. Thank you for your suggested edit!
>
> ## Questions
> 1. You are correct that using IQR to normalize performance will be a lower variance estimate. We agree it may be more appropriate, especially when practitioners do not wish to run 200 seed experiments. We will add results that use IQR normalization.
> We have used the term “run” synonymously with seed. Given that the goal of RL is to maximize expected return, the best seed will not be an informative alternative to mean performance across seeds.  We assume that you meant something different with your use of the word run. Perhaps you meant final policy performance, as mentioned before? If so, we would happily include the results with the final performance in the appendix.
>
> 2. Yes, we can improve the discussion and conclusions. We will note in the discussion how the Mujoco environments have similar observations, reward scales, etc., and our results do not generalize to other classes of environments. Moreover, we will emphasize that PPO and Mujoco were chosen because they are heavily studied, and more importantly, they allowed us to demonstrate the metrics can still provide insight in these domains.
>
> ## Limitations
> The presented methodology is:
> 1. Collect data about the performance of different algorithms concerning their hyperparameters across different environments
> 2. Normalize performance (via min-max, CDF, or some other way)
> 3. Using the data that has already been collected and normalized to compute sensitivity and effective hyperparameter dimensionality as we defined.
>
> We are agnostic to how practitioners collect their data. Grid search will not scale to larger hyperparameter spaces, so something like a random search may be required and fits the proposed methodology.  At the same time, please note that RL is incredibly noisy.  There are many cases where studies with few seeds have greatly misled the community: https://arxiv.org/pdf/2304.01315
>  Statistical estimates require many samples; as you correctly noted, we use min and max for normalization, which are high variance. It made sense to use the available computation that we had to run more seeds instead of additional environments, algorithms, etc.

---

> > ### Comment · Reviewer_ydma · 2024-08-08
> > **A promising paper that needs more polish**
> >
> > I thank and commend the authors for the strong points laid out in the global response, the revision of their figures, and their direct comments. I find figure 2 in the additional pdf (global response) extremely interesting and am surprised by the extreme contrast in results when comparing figure 4 (main paper) with figure 1 (pdf rebuttal).
> >
> > ---
> > I agree with the point that mujoco/ brax was only a test-set to showcase your visualization method, and I did not find any overly strong claims or conclusions in these sections in particular. However, the main problem that I have is that despite the costs of the experiments, and the proposed visualization, I can still not draw any insights from these results. Which is unfortunate.
> >
> > The issue that I had is that this section has too strong of a focus on the results, for example, section 4.2 is a lengthy discussion of most parameters (the details of which, I believe were a bit beside the point) and section 4.3 simply enumerates everything we can see in figure 4.
> >
> > I want to see a critical discussion of: how can we use this visualization method to gather insight and what can we do with this information. This should be accompanied by a well executed test-case. This should include multiple ways to look at the same data (different normalization, environment subsets, conditional parameters, or more). Like I mentioned before, I think figure 2 of the rebuttal-pdf adds tremendously to this discussion.
> >
> > ---
> > Considering all my previous points, I still think this paper lacks polish. It looks promising, but it is not quite there yet. I will raise my score from 3 to 4 and contribution from 1 to 3, since the authors did address many concerns.

---

> > > ### Author Response · Authors · 2024-08-13
> > >
> > > Here are two insights that can be drawn from Figure 2 of the pdf rebuttal.
> > >
> > > While it has been known in the literature that observation normalization matters a lot for performance, we can now see that the performance gain comes with, or is even partially enabled by, increased sensitivity.
> > > It has also been previously reported that advantage normalization does not matter for performance. This appears to be true. However, it somewhat lowers sensitivity while retaining similar performance, making it possibly more interesting for practitioners than the literature has presented it.
> > >
> > >
> > > Thank you for pointing out that section 4.3 could be deeper. For the camera-ready, we will drastically shorten section 4.2 (or move it to the appendix) and expand section 4.3 to include additional discussion and figures (different normalization schemes, environment subsets, etc.) as discussed in the rebuttal and responses.

---

### Official Review · Reviewer_aSZK · 2024-06-26

**Soundness:** 2
**Presentation:** 3
**Contribution:** 3
**Rating:** 7
**Confidence:** 4

**Summary:**

The paper proposes a method to analyse how sensitive are RL methods with respect to hyperparameter tuning. The author argue that one method may perform well on average but require more HPO tuning per task which hides some computation and prevent from having comparable results. They introduce a sensitivity metric which measures the difference between the best hyperparameter tuned per task and the performance of the best hyperparameter on average. Then, they propose a quadrant analysis where both the sensitivity and the performance of a tuned score are displayed which allows to compare algorithms on those two dimensions. Experiments are then performed on PPO normalization variants where they show that current normalization have different trade-offs: some approaches improves the scores but increase hyperparameter sensitivity while other conversely lower scores and sensitivity but no approach allows to improves both scores and sensitivity currently. Finally, the authors study how many hyperparameters require tuning while having the other fix and while keeping 95% of the best performance.

**Strengths:**

* The paper is very well motivated. It tackles an important problem for RL (and in general on how to account for HPO sensitivity when reporting results.
* The paper is very well written and easy to follow, the methods are well described and the experiment are very sound
* The paper would provide a valuable contribution if the results of the runs are released (will they?, see my question)

**Weaknesses:**

The paper is currently missing some analysis regarding the method stability (see questions). For instance how much the proposed method would be stable and reliable with respect to different environments (does adding one environment change the results completely?) or normalization (e.g. using CDF instead of min max).

**Questions:**

Here are points that would be important for me to raise my score:

* In figure 4, are the results stable if you leave one out one environment? If you compute the plot 5 times each time leaving one of the environment, do you have similar plots? If not, then the analysis on the paper will be less warranted (that such and such method is more stable with hyperparameter).
* Will the paper release the dataset of evaluation in addition to the code? (the code would have limited interest compared to the data) I would highly encourage to share the dataset, possibly with a script to reproduce some of the paper figures (for instance as was done by https://github.com/google-research/rliable) The dataset would be also very useful to simulate and compare HPO methods, in particular if it contains the evaluation per iteration (and per 100s of iterations for instance).

* Figure 4: there are very large outliers for the obs zero mean normalization sensitivity and the distribution is very skewed, could you explain why they happen?

* One popular method is f-Anova to study hyperparameter importance, could you include an analysis using it? It would be useful for practitioners to know which HP are most useful.

* In the limitation section, you mention that the results may change under a different normalization (say CDF instead of min-max). This is indeed an important point, could you report the result for Fig4 as well with CDF? I assume it should be a minor change as it is just changing the metric. However, it seems important to assess how much the method would be impacted by different normalization (even if the results change, the paper would still be valuable)


Here are points that are more details:
- You have l110 p(w, e, h ,\kappa) and then p(a, e, h), it would be nice to unify the notation
- l151: I think it would be useful to precise that the quantity is always >= 0
- "The shaded region is a 95% Student t-distribution confidence interval around the mean return over 200 runs "
=> what is the CI not covering the mean in walker 2D?

**Limitations:**

Yes

---

> ### Author Rebuttal · Authors · 2024-08-07
>
> First, thank you for your review and comments on improving our work. We will address your specific questions here and focus on general concerns shared between the reviews in our general rebuttal.
>
>
> ## Weaknesses
> As requested, we have included 5 repeats of the Figure 4 plots, leaving out an environment each time in the PDF attached to the general rebuttal. In addition, another reviewer asked us to demonstrate results with (p,1-p) percentile normalization instead of min-max; we will perform this and include these results.
>
> ## Questions
> 1. We have included a figure in the PDF attached to the general rebuttal that does this. Leaving out environments does shift the reference point, and the position of the variants shifts somewhat relative to the reference point. However, the regional position of the variants is mostly consistent (e.g., observation ZM normalization is still more performative and sensitive than the reference point).
>
> 2. Yes, we will do this. The code for reproducing the plots was included in the supplementary material, but it is somewhat useless without the data. We will look into providing the data through Google Cloud or some other platform.
>
> 3. Digging deeper into this, we found a minor error in the plotting code. After re-running experiments, we found that it did not impact the main message of our paper or the vast majority of the particular empirical outcomes. The corrected Figure 4 will be included in the PDF for the general rebuttal.
>
> 4. The focus of our study has been characterizing algorithms’ performance across sets of environments with respect to their hyperparameters. The work we have seen that uses f-Anova has focused on characterizing the importance of hyperparameters within an environment, an important but slightly tangential goal. We could report the hyperparameters that were most impactful to tune per environment (measured when creating Figure 5) and report how the important hyperparameters and their values change when tuning pairs of hyperparameters jointly, etc. Do you believe this would be valuable?
> 5. Another reviewer suggested normalizing by a (p,1-p) percentile range instead of min-max, which has a lower variance. We will include this normalization and compare it with min-max in the final version.
> As for CDF normalization, after further consideration, we are unsure if it makes sense to visualize it in the style of Figure 5. Taking an expectation through a non-linear operator does not preserve orderings.  We normalize the AUCs of runs (seeds) and then average across runs to obtain the expected normalized performance. The interpretations of regions in Fig 4 may have very different meanings under CDF normalization. An algorithm could have a worse average return but be higher on the CDF normalized Figure 5 plot. E.g. some seeds perform slightly better in units of return, but are placed much higher in CDF from the nonlinearity. Nevertheless, it is a minor change and we can include results with CDF normalization in the appendix if requested.
>
> ## Detailed comments
> 1. Thank you for catching the notation error. It will be fixed for the camera-ready.
> 2. We will clarify this. Another reviewer suggested noting the connection to Jesen’s inequality. This will be clarified for the camera-ready version.
> 3. Thank you for catching this. After re-running the code and examining it, the confidence interval fully covers the mean. We suspect that a post-processing formatting issue introduced this error in the document. The corrected version is included in the general rebuttal PDF, and the figure will be corrected for the camera-ready version.

---

> > ### Comment · Reviewer_aSZK · 2024-08-12
> > **answer to rebuttal**
> >
> > Thank you for your answer.
> > 1. Thanks for adding this figure, I think it is important to verify this stability to make sure the statement of the papers are applicable in generalized settings.
> > 2. Great to hear, as mentioned in my review, this data can be valuable for other research.
> > 4. The analysis you suggest makes sense and would provide value (I gave f-anova as an example but any analysis on hp importance would be valuable). I agree with you that studying hyperparameter importance is tangential to the main point of the paper but still seems related (it may be that some algorithms have the same hyperparameters to tune and other requires a larger set which is also interesting for practitioners, in addition to know which hyperparameters had the most effect)
> > 5. I get your point about CDF, I dont mind if you use (p, 1-p) percentile transformation instead as long as you make sure that the results are not completely tied to one normalization. Regarding the downsides of the CDF you mention, I think those are standards points not necessarily tied to your use-case: the CDF gets rid of the scale (only the ordering matter), this has benefits (robustness to outliers, uniform distribution obtained) and downsides (sometimes the scale is important) which is why this normalization is good to have in addition to min-max as it has different trade-offs.
> >
> > I have raised my score given that some of my points were addressed.

---

> > > ### Author Response · Authors · 2024-08-13
> > >
> > > We appreciate your comments and thank you again for your thoughtful review.

---

### Official Review · Reviewer_yuTN · 2024-06-29

**Soundness:** 2
**Presentation:** 2
**Contribution:** 2
**Rating:** 4
**Confidence:** 3

**Summary:**

This paper proposes a new evaluation regime for reinforcement learning. As opposed to only taking into account benchmark performance (i.e., final return), as is ocmmon in previous literature, this work suggests considering an extra dimension of how sensitive algorithms are to hyperparameters in tuning based on a heuristic developed in the paper. They analyse PPO, and some of its variants, using this new evaluation regime, and find that performance increases often correspond to increased sensitivity to hyperparameters. Finally, the paper considers a top-k approach to hyperparameter tuning, adjusting only the most impactful hyperparameter, to see how the baseline algorithms compare under a more limited hyperparameter tuning regime.

**Strengths:**

- The paper clearly takes care to use an extensive number of experiments, possibly reaching into the realm of unnecessary, to provide some rigour to their results.
- The domain being considered - considering the hyperparameter sensitivity of algorithms in addition to their performance - is an underexplored area and one which becomes increasingly important as the cost of experiments increase.
- Approaching this problem in a visual setting seems reasonable.
- I like the approach of distilling hyperparameter optimisation to optimising only a smaller set of more important values; this has real benefits in enabling tuning of the majority of hyperparameters in *cheap* environments and tuning the key values only in more expensive environments.

**Weaknesses:**

- Noting appendix A (the table of how many hyperparameters each algorithm has); the definition of 'hyperparameters' seems pretty weak, and a lot of those included seem to just be design decisions of the actual algorithms.
- I find reading the plots quite confusing, exacerbated by all of the different colours marking each of the areas. I think this obfuscates the message of the plots and as a reader makes it hard to come to conclusions.
- Given this work is based on PurejaxRL, which I think runs on Brax for 1e8 frames, these experiments seem very short (3e6 frames) and possibly doesn't give the algorithms the full opportunity to converge.
- There is very limited discussion of preexisting literature in this space. While I appreciate this takes a subtly different tack (calling for us to measure how sensitive to hyperparameters RL algorithms are, rather than designing algorithms with few hyperparameters), I would expect to see significant more discussion of AutoRL literature. Framing this work better in related work would definitely strengthen the paper.
- The formatting looks quite off with the figures. I think this is because a lot of the captions are to the side of the figures, rather than below.
- Considering the actual hyperparameters being tuned, the results feels slightly disingenuous. If we assume the practitioner running the hyperparameter tuning, we would possibly expect them to select more reasonable values or focus the search in a significantly more targeted way. For instance, it is no surprise that the highly performant methods saw large performance decreases (i.e., hyperparameter sensitivity) when evaluated with entropy coefficients in the range [0.001, 10] or learning rates spanning 5 orders of magnitude. Instead, it would be much more sensible to explore how the performance changes with reasonable hyperparameters that one is more likely to practically tune over (i.e., we generally have a good idea of where to start with values, so are unlikely to try an entropy coefficient of 10).
- It would be good to see some exploration of algorithms which have done something different than just considering the changes to normalisation in PPO returns or observations. It feels like some broad statements are made despite the fact that this analysis was only taken in a single setting.

**Questions:**

- In the plots, the authors state that they plot 95% confidence intervals over 200 seeds; but in the y-axis, all of the 'confidence intervals' are only negative. I am not particularly clear why. Is this a mistake, or am I misinterpreting results? At the same time, are the circles representing the mean score over runs, environments, or both; and therefore, are the confidence intervals defined over the runs, the enviroments or a big stack of (in this case) 1000 results per point? Is comparison for the hyperparameter sensitivity done per seed or for the averages over seeds? I think there's a lot of questions about how exactly these results are being presented which doesn't come through in the paper.
- In some cases, increased hyperparameter sensitivity can be a positive - it can give us extra opportunities to boost performance. Do you think this should be mentioned as a limitation of this method, which effectively calls for less sensitivity?
- Why did you use 200 seeds per hyperparameter tuning configuration, which is significantly greater than would be used in most other research? In practice, would it not have been more sensible to have used a much smaller number of seeds (eg 8), and focused on a much tighter range of hyperparameters?
- As a brief aside, the authors have not removed the instructions of the Neurips checklist as requested, which should be done if the camera ready copy is released.

**Limitations:**

The authors briefly discuss some limitations of their methods; in particular, that their proposed metrics are highly environment specific (even in this case, they are looking at only one suite of environments which may end up biasing results), and will need reevaluation every time you shift to a new set of domains. However, I think there are some other key limitations that are not discussed, many of which have been raised above. In particular:
- In addition to being environment specific, the metrics proposed are going to be very specific to the range of values which are tested. I have mentioned above how the values chosen for hyperparameter tuning here are not particularly reasonable.

---

> ### Author Rebuttal · Authors · 2024-08-07
>
> Thank you for your review. We appreciate your feedback.
>
>  ## Weaknesses
>
> 1. We recognize that the delineation between hyperparameters and algorithm design can be fuzzy. We cite the sources used for our counts and, when possible, try to stick to hyperparameter lists identified by the original papers (e.g., the DQN Nature paper has a table of 16 hyperparameters, so the value in our table is 16).  When this was not possible, as a guiding principle, we tried to identify parameters of the algorithm that could easily be modified and may contribute to performance gains if tuned per environment. We did not count parameters related to network architecture (e.g. number of neurons in hidden layers).
>
> 2. The two-dimensional plot is motivated by the fact that nuance is lost when algorithms are evaluated using one metric. Studying the relationship between sensitivity and performance can aid insight. Perhaps there is a better way to discriminate between regions other than colors. We would be happy to try out any suggestions the reviewer has.
>
> 3.  The Mujoco experiments that we have observed in the literature ran for between 3e^6 - 5e^6 steps (e.g., OpenAI has published a 3e^6 step benchmark for the 5 environments that we tested: https://spinningup.openai.com/en/latest/spinningup/bench.html. Note we are evaluating based on AUC and not final performance; for final performance one might choose to run longer.
>
> 4. You are correct that we are addressing a different problem. Often, algorithms in the AutoRL literature aim to tune hyperparameters via some HPO process (which sometimes contain additional hyperparameters of their own).  Future studies could use the definitions provided here to try to understand if the algorithms proposed in the AutoRL literature reduce sensitivity over the base algorithms that are being modified. We are proposing an evaluation methodology, not a solution to hyperparameter tuning.
> 5. We will improve formatting.
> 6. It is not uncommon to sweep stepsizes over multiple orders of magnitude see https://arxiv.org/pdf/2407.18840, which considers actor step size as low as 10^{-6} and as high as 10^{-2}). While it may seem a bit unusual to try an entropy coefficient of 10, consider that the proper order of magnitude of the entropy coefficient will depend heavily on the scale of the advantage term in the actor loss. This will be affected by the type of advantage normalization performed or the order of magnitude of the reward function if the advantage is left unnormalized. Existing literature (Table 4: https://arxiv.org/pdf/2306.01324 from the AutoRL literature) has already searched spaces of 4 orders of magnitude to find entropy coefficients for PPO in Brax. Considering that we are ablating various forms of normalization across different sets of environments, it does not seem unreasonable to try 5 different orders of magnitude. In our results, the entropy coefficient of 10 was not bad in every environment. In fact, there was a hyperparameter setting with an entropy coefficient of 10 for the observation normalization PPO variant that was in the 97th percentile of the settings we tried for Walker2d. Nevertheless, please note this paper's main contributions are the sensitivity and dimensionality metrics and the plots for visualizing them. The PPO study is used to demonstrate how these tools might be used.
> 7. It would be great to apply the sensitivity and effective dimensionality metrics to other algorithms, environment distributions, etc. Note that claims will always be limited to some small set of agents and environments (as you noted above); general claims are never possible, especially considering many RL benchmark environments have been developed specifically to test current algorithms. We were very careful not to make claims beyond the methods tested. If there are specific statements in the paper that you feel are overreaching, let's discuss them.
>
> ## Questions:
> 1. The confidence intervals were computed by creating bootstrap datasets, i.e., sampling seeds with replacement from the original dataset and computing the performance and sensitivity metrics on the bootstrap datasets. This creates distributions of algorithms' performance and sensitivity scores. The upper and lower endpoints of the confidence intervals are the 97.5th and 2.5th percentiles of the distributions. Please note that the collapse of the upper endpoint of the performance CI in Figure 5 resulted from a minor bug in the plotting code. We have re-run the bootstrap and included the corrected figure 5 in the attached PDF.
>
> 2. There are cases where a practitioner would be okay with increased sensitivity, especially if there are even larger increases in performance. The question highlights the benefit of studying the performance-sensitivity relationship of algorithms on a 2d plane, such as in Figure 4
>
> 3. Because our proposed metrics are nonstandard, we wanted to avoid statistical tests that use distributional assumptions (e.g., student-t assumes normality). This is why we chose to use percentile bootstrap. However, percentile bootstrap CIs are often quite wide; therefore, we chose to run many seeds to obtain useful CI estimates. There is a long line of work suggesting the current practice of fewer than 10 seeds is not enough:
> https://arxiv.org/abs/2304.01315,
> https://arxiv.org/abs/1904.06979,
> https://openreview.net/pdf?id=Xe7n2ZqpBP,
> https://rlj.cs.umass.edu/2024/papers/RLJ_RLC_2024_330.pdf
>
>
> ## Limitations:
> 1. The computed sensitivity metric is intimately tied to the distribution of environments used, the process for choosing hyperparameter values, the number of seeds, etc.  An underlying assumption is that a practitioner cares about evaluating algorithms with respect to some distribution of environments. If the practitioner is consistent in their data collection process, these metrics can be used. As argued above, respectfully, we feel the ranges used were appropriate given the algorithmic variations under study.

---

> > ### Comment · Reviewer_yuTN · 2024-08-08
> >
> > Thank you for extensive response to my review.
> >
> > A few thoughts are below. Where satisfied, I have not included response to each point for brevity's sake.
> >
> > # 2D plot
> >
> > I think the key for me about the colouring is that it enforces a fairly arbitrary segmentation that I think gets in the way of analysis. I think, personally, I would prefer to see results simply plotted on these axes, with analysis separated into text.
> >
> > # Comparison with AutoRL
> >
> > While I agree there is a difference, I think the takeaway is that both AutoRL and your work have similar motivations - to promote systems where an environment is put in and a policy comes out without human input. As such, I still think this is worthwhile comparison. In a sense, you can consider that AutoRL algorithms are 'hyperparameter free' in that they deal with hyperparameters internally, and thus have no hyperparameter sensitivity.
> >
> > ---
> > In addition to the above, I still feel there is a missing component considering how human-in-the-loop and prior work would focus a lot of these search efforts by offering intuition about the kind of hyperparameter ranges which are useful. That said, I am so far satisfied that some of my concerns have been addressed sufficiently, and thus have increased my score from 3 to 4. I remain open to discussion about the above.

---

> > > ### Author Response · Authors · 2024-08-13
> > >
> > > 2D Plot
> > > While color may not be the best choice for visualizing these segmentations, we don’t believe the segmentation is arbitrary. Each of the segmentations has a different interpretation of its relation to the reference point. The slope one line passing through the reference has unique importance as it marks the points where observed performance gains are directly attributable to per-environment hyperparameter tuning.
> > >
> > > Comparison with AutoRL
> > >
> > > We agree that, like AutoRL, we are interested in promoting methods that require less human intervention and tuning to apply. The key difference we see between our work and AutoRL is that we are demonstrating the utility of our methodology with an experiment on PPO, not proposing a new way to tune hyperparameters in RL. While AutoRL methods tune an RL algorithm's hyperparameters internally, it should be noted that the AutoRL algorithms themselves often have hyperparameters (e.g., the scaling factor and crossover factor parameters in DEHB). For the camera-ready, we will include an additional discussion of the AutoRL literature and how the AutoRL community could use our method to measure their effectiveness at improving performance while reducing (hyper-)hyperparameter sensitivity.
> > >
> > > Thank you for your additional comments. We appreciate your feedback and discussion.

---

### Official Review · Reviewer_6Bue · 2024-07-01

**Soundness:** 3
**Presentation:** 3
**Contribution:** 3
**Rating:** 7
**Confidence:** 4

**Summary:**

The paper introduces an empirical framework for assessing the hyperparameter sensitivity of reinforcement learning algorithms.
The framework consists of two metrics: 1. hyperparameter sensitivity, which gives a normalized difference in performance between the per-task best hyperparameters and the across-task best hyperparameters. 2. effective hyperparameter dimensionality, which gives the number of hyperparameters that can be left the same as the across-task hyperparameters while tuning the rest and still obtain a threshold say (95%) of the per-task best hyperparameter performance.
Using these metrics in addition to performance metrics, practitioners and researchers can have a better idea of the benefits/downsides of a modification to an algorithm (how much better vs how much more sensitive)
The framework is used to compare several normalization variants of PPO introduced in the past years on the continuous control environments, giving a better picture of their contribution.

**Strengths:**

The problem is well-motivated. Hyperparameter sensitivity is a well-known issue in deep RL with novel algorithms often providing better performance at the expense of a higher sensitivity. Such tradeoff must be made explicit.
The framework introduced in the paper provides an effective way to draw that tradeoff with a clear interpretation of it using Figure 3.

The metrics are simple and quite natural and provide a solid starting point for a hyperparameter sensitivity framework.
Although computationally expensive and still to prove if its results transfer across domains, the framework is likely to have a high impact on the field.
At least on the continuous control with Brax and PPO variants where the framework has been used as an example.
In particular thanks to the thorough and accurate experimental protocol, with 200 seeds per experiment and confidence intervals.

**Weaknesses:**

Metric definition:
- The effective hyperparameter dimensionality depends on the total number of hyperparameters of an algorithm and is likely to scale with it, so this makes it incomparable between algorithms with a different number of hyperparameters. Perhaps counting the number of parameters that changed instead would make it more comparable across algorithms, as is what would dictate the budget of practitioners eventually.


Hyperparameters:
- I would consider the minibatch size and the number of epochs in PPO to be critical hyperparameters as well. It's not clear how the choice of hyperparameters to sweep over was made.
- (minor) The epsilon in the denominator of the minibatch normalization may also play a big role.

Inconsistent/confusion notation:
in line 91, the tuple (w,h) defines an agent (a,), but this is used in a confusing and inconsistent way throughout the paper.
- line 116 $\hat p(a, e, h): if $a$ is there then $h$ is redundant. It seems like $a$ there stands for $w$.
- Equation 1 same. The $a$ should be a $w$.
- Equation 2 and 5:  2 and 5: $\Gamma(a, e, w)$ not it's even more confusing there is both an $a$ and an $w$ and there is an $h$ missing.

Claims:
Line 321 "vastly difference effective hyperparameter dimensionalities": the largest gap observed in the paper is from 2 to 4, so I would not extrapolate here, though arguably that's indeed an additional 2 dimensions to sweep over so scales exponentially.

**Questions:**

Reporting the mean under the curve (AUC) with 95% bootstrapped confidence intervals over 200 runs is great, but I would expect some discussion on the use of a confidence interval, as it collapses the shaded area across the 200 runs to the statistic being computed (here the mean AUC).
To me, using a dispersion measure like the variance of the mean AUC would also be valid, as some hyperparameters would have more variance than others (although this could also be seen as an additional dimension of sensitivity).
I would appreciate it if the authors could comment on this.
Also, at what point are the confidence intervals computed? When computing the expected performance or when computing the hyperparameter sensitivity, etc?

**Limitations:**

The authors adequately state the limitations of their work and its broader impacts. In particular the potential limitation of the framework to a specific environment distribution.

---

> ### Author Rebuttal · Authors · 2024-08-07
>
> Thank you very much for your review. Your comments are appreciated for improving this work.
>
> ## Weaknesses
> Metric definition:
>
> The effective hyperparameter dimensionality already does this by counting the number of the hyperparameters necessary to obtain a large fraction of peak performance. If algorithm x has two very sensitive hyperparameters that need to be tuned per environment, and algorithm y has 40 hyperparameters, but only one of them is sensitive, where the others are okay being left to default, then algorithm x will have an effective hyperparameter dimensionality of 2 while algorithm y will have an effective hyperparameter dimensionality of 1.
> Of course, as the number of hyperparameters grows, its number of sensitive hyperparameters may also grow, but this does not make it incomparable to other algorithms.
>
> ## Hyperparameters
> Thank you for the suggestions! Based on our review of the literature and previous experience, we chose hyperparameters that appeared to be important. The hyperparameter choices you suggested would be very interesting to investigate. If you believe it would add to the paper, we would be willing to run additional experiments with batch size, epochs, etc., before the camera-ready deadline.
>
>
> ## Inconsistent notation
> This resulted from a notation change for the algorithm set. Thank you for finding the inconsistency; it will be fixed for the camera-ready.
>
> ## Questions
> The confidence interval is computed by creating bootstrap datasets, i.e., sampling seeds with replacement from the original dataset, and then computing the performance and sensitivity metrics on the bootstrap datasets. This creates distributions of the algorithms' performance and sensitivity scores. The upper and lower endpoints of the confidence intervals are the 97.5th and 2.5th percentiles of the distributions. Please note that the collapse of the upper endpoint of the performance CI in Figure 5 resulted from a minor bug in the plotting code. After re-running experiments, we found that it did not impact the main message of our paper or the vast majority of the particular empirical outcomes. We have included the corrected Figure 5 in the PDF attached to the general rebuttal.

---

> > ### Comment · Reviewer_6Bue · 2024-08-11
> >
> > I thank the authors for the clarifications.
> >
> > Regarding the bootstrapped confidence intervals, what I mean is that the more samples available and the more points drawn at a time the smaller the interval will be.
> > This is different from a dispersion measure, like the variance, which would not shrink with more data.
> >
> > I maintain my score.

---

> > > ### Author Response · Authors · 2024-08-13
> > >
> > > Thank you for your response and for clarifying. We are sorry that we missed this point in our initial response. We used a confidence interval as we are interested in understanding the exact sensitivity and performance under different algorithmic variations and so plot the confidence in our sample estimates.  We will include a table in the appendix that reports the standard deviation across seeds of the AUC for each hyperparameter setting tested in each environment.

---

### Author Rebuttal · Authors · 2024-08-07

Thank you to the reviewers for your questions, comments, and suggestions on improving our work! We appreciate your feedback. We have addressed specific comments in the individual rebuttals and will provide a general response to comments shared among the reviewers here.

The main contributions of our paper are the hyperparameter sensitivity and effective hyperparameter dimensionality metrics and the plots for visualizing them. There is confusion that because our study performed many runs with a grid search this is what we are advocating for. Our methodology does not require one to do a grid search, 200 seeds, AUC performance metrics, min-max score normalization, or to perform a percentile bootstrap. The hyperparameter sensitivity and effective hyperparameter dimensionality metrics are agnostic to these choices. If a practitioner made a different empirical design choice than us on one of these decision points, then they may very well see different results; what is most important is that they are consistent and fair when comparing algorithms in their study. For example, another empiricist could do a random search with 10 seeds per hyper-parameter setting, consider final policy performance, use a different score normalization technique, report standard errors, and still use the definitions and plots that we provide.

Several reviewers correctly noted that the sensitivity metric is intimately tied to the environment set and that our findings are thus limited to Brax Mujoco environments tested. We acknowledge this. No matter how large of an environment set is chosen, this will always be true; regardless of whether we include Atari, Minecraft, DMLab, or any other suite of environments, empirical results will not allow us to make claims about environments outside the evaluated environment distribution. What is being missed here is an important finding that even when there is only one common environment suite, sensitivity emerges and is a big deal. Limiting ourselves to Mujoco was a feature, not a bug of our study—notwithstanding the fact that these Mujoco environments are widely used to rank and evaluate new algorithms.

Several questions were raised regarding various empirical design choices. We addressed each question in the per-review rebuttals; however, we will also summarize our justifications for them here.

 The 200 seed bootstrap CI was performed for the following reasons: min-max estimation is high variance, the sensitivity metrics we propose are nonstandard, we did not wish to rely on confidence intervals that make distributional assumptions, there are many forms of stochasticity in RL, and existing literature has demonstrated that studies with small numbers of seeds can be misleading. We dug deeper based on the interesting questions from the reviewers. It revealed a minor error in the bootstrap CI code that did not impact the main messages of the paper or the vast majority of the particular empirical outcomes. The good news is that after running our experiments again with the bug fixed, two things that did change helped answer two of the questions from the reviewers regarding the size of the confidence intervals.  Please see the attached PDF for the corrected Figure 4.

As for the grid search, we believe the values chosen are appropriate based on existing literature:  https://arxiv.org/pdf/2306.01324, https://arxiv.org/pdf/2407.18840, and the argument presented in our rebuttal to reviewer YuTN.

The performance metric AUC was chosen as it captures the agent’s rate of learning. Additional studies using other performance metrics may be valuable. We will add results with the final policy performance used as a metric to the appendix as requested by reviewer ydma.

The choice of min-max normalization is indeed high variance. An alternative (suggested by ydma) would have been to normalize by a (p,1-p) percentile range (such as IQR). We will include results with this type of normalization.


Finally, reviewers noted some notation typos, sentence errors, and suggestions for added discussion. Thank you very much for pointing them out! We will ensure that all of these edits are made for the camera-ready submission. We appreciate your questions, comments, and criticisms for improving this research.

---

### Decision · Program_Chairs · 2024-09-25

**Decision:**

Accept (poster)

**Comment:**

This paper tackles a very important problem in RL: how do we know when a newly proposed approach leads to generalizable improvements. The reviewers agree that it is very well-motivated and the proposed metrics are simple and intuitive. However, they have concerns about the robustness of the results due to the selection of environments only from Mujoco, a fixed set of hyperparameters, and the use of the min/max in the environment normalization score. Nevertheless, I feel that the proposed framework is valuable and can potentially impactfully improve evaluation in RL.